# FMNL2 regulates actin for endoplasmic reticulum and mitochondria distribution in oocyte meiosis

**Meng-Hao Pan[1,2], Kun-Huan Zhang[1], Si-Le Wu[1], Zhen-Nan Pan[1], Ming-Hong Sun[1], Xiao-Han Li[1], Jia-Qian Ju[1], Shi-Ming Luo[3], Xiang-Hong Ou[3], Shao-Chen Sun[1]***

[1]College of Animal Science and Technology, Nanjing Agricultural University, Nanjing, China; [2]College of Veterinary Medicine, Northwest A&F University, Shaanxi, China; [3]Fertility Preservation Lab, Reproductive Medicine Center, Guangdong Second Provincial General Hospital, Guangzhou, China

**\*For correspondence:**
sunsc@njau.edu.cn

**Competing interest:** The authors declare that no competing interests exist.

**Abstract** During mammalian oocyte meiosis, spindle migration and asymmetric cytokinesis are unique steps for the successful polar body extrusion. The asymmetry defects of oocytes will lead to the failure of fertilization and embryo implantation. In present study, we reported that an actin nucleating factor Formin-like 2 (FMNL2) played critical roles in the regulation of spindle migration and organelle distribution in mouse and porcine oocytes. Our results showed that FMNL2 mainly localized at the oocyte cortex and periphery of spindle. Depletion of FMNL2 led to the failure of polar body extrusion and large polar bodies in oocytes. Live-cell imaging revealed that the spindle failed to migrate to the oocyte cortex, which caused polar body formation defects, and this might be due to the decreased polymerization of cytoplasmic actin by FMNL2 depletion in the oocytes of both mice and pigs. Furthermore, mass spectrometry analysis indicated that FMNL2 was associated with mitochondria and endoplasmic reticulum (ER)-related proteins, and FMNL2 depletion disrupted the function and distribution of mitochondria and ER, showing with decreased mitochondrial membrane potential and the occurrence of ER stress. Microinjecting *Fmnl2-EGFP* mRNA into FMNL2-depleted oocytes significantly rescued these defects. Thus, our results indicate that FMNL2 is essential for the actin assembly, which further involves into meiotic spindle migration and ER/mitochondria functions in mammalian oocytes.

## eLife assessment

This study presents **useful** findings regarding the role of formin-like 2 in mouse oocyte meiosis. Some of the data are supported by **incomplete** methodological details and analyses, and several conclusions are overstated. This paper would be of interest to reproductive biologists.

## Introduction

Mammalian oocyte maturation is an asymmetric division process that generates a large egg and a small polar body. This asymmetry is critical for the following fertilization and early embryo development. After germinal vesicle breakdown, the meiotic spindle is organized at the center of the oocyte, and then it migrates to the oocyte cortex at the late metaphase I (MI). The oocytes are arrested at metaphase II (MII) after the extrusion of first polar body (***Pan and Li, 2019***; ***Yi et al., 2013a***). Actin filaments, as the most widely distributed cytoskeleton in cells, regulate various dynamic events during oocyte meiotic maturation (***Sun and Schatten, 2006***), and two key events are the spindle migration and cortical reorganization in mammalian oocytes (***Duan and Sun, 2019***; ***Sun and Kim, 2013***; ***Yi et al.,***

*2013a*). Small GTPases and actin nucleation factors are shown to promote the assembly and function of actin. The actin nucleation factors are the molecules that directly promote the actin assembly: Arp2/3 complex control the assembly of branched actin, and formin family member Formin2 (FMN2) and Spire1/2 control the assembly of linear actin. These proteins are all proposed to play a role in actin-related spindle migration and cytokinesis during mammalian oocyte maturation (*Li et al., 2008*; *Sun et al., 2011*; *Yi et al., 2013b*). The cortex protein Arp2/3 complex nucleates the actin to produce a hydrodynamic force to move the spindle toward the cortex, and regulates cytokinesis during oocyte maturation (*Sun et al., 2011*; *Yi et al., 2013a*). FMN2 and Spire1/2 nucleates actin around the spindle in the cytoplasm to give the meiotic spindle an initial power for migration (*Li et al., 2008*; *Pfender et al., 2011*).

Besides Formin2, the DRFs (diaphanous-related formins) subfamily in the formin family has been extensively studied. The DRFs family consists of mDia, Daam, FHOD, and FMNLs (*Kühn and Geyer, 2014*). The 'Formin-like' proteins (FMNLs) subfamily includes FMNL1 (FRL1), FMNL2 (FRL3), and FMNL3 (FRL2). Like other Formin family proteins, FMNLs play important roles in cell migration, cell division, and cell polarity (*Katoh and Katoh, 2003*; *Kühn and Geyer, 2014*). While FMNL2 is widely expressed in multiple human tissues, especially in the gastrointestinal and mammary epithelia, lymphatic tissues, placenta, and reproductive tract (*Gardberg et al., 2010*). As an important actin assembly factor, FMNL2 accelerates the elongation of actin filaments branched by Arp2/3 complex (*Kage et al., 2017*). In invasive cells, FMNL2 is mainly localized in the leading edge of the cell, lamellipodia and filopodia tips, to improve cell migration ability by actin-based manner (*Block et al., 2012*; *Kage et al., 2017*; *Zhu et al., 2011*). FMNL2 is also involved in the maintenance of epithelial–mesenchymal transition in human colorectal carcinoma cell (*Li et al., 2010*). Besides its roles on the actin assembly, emerging evidences indicate that FMNL2 may interact with organelle dynamics. It is shown that FMNL2 is related with the Golgi apparatus, since the absence of FMNL2/3 can cause the Golgi fragmentation (*Kage et al., 2019*). However, till now the roles of FMNLs especially FMNL2 on oocyte meiosis are still largely unknown.

In the present study, we disturbed the FMNL2 expression and explored the roles of FMNL2 during mouse and porcine oocyte meiosis. Our results showed that FMNL2 was essential for the polar body size control and successful extrusion; and these abnormal phenotypes might be due to aberrant actin-based meiotic spindle migration. Meanwhile, we also found that FMNL2 was essential for the functions and distribution of mitochondria and endoplasmic reticulum (ER). Therefore, this study provided the evidence for the critical roles of FMNL2-mediated actin on spindle movement and organelle dynamics in mammalian oocytes.

## Results

### Expression and subcellular localization of FMNL2 during oocyte maturation

To examine FMNL2 expression and localization in mouse oocytes at different stages, western blotting, mRNA microinjection, and immunofluorescence staining were performed on freshly isolated germinal vesicle (GV)-stage oocytes and oocytes cultured for 4, 8, and 12 hr, corresponding to germinal vesicle breakdown (GVBD), MI, and MII stages, respectively. The results indicated that FMNL2 all expressed in GV, MI, and MII stages during mouse oocyte maturation (GV, 1; MI, 0.82 ± 0.07; MII, 0.61 ± 0.10, *Figure 1A*). As shown in *Figure 1B*, *Fmnl2-EGFP* mRNA microinjection showed that FMNL2 accumulated at the oocyte cortex during the GV, GVBD MI, and MII stages. Besides, FMNL2 also localized at the spindle periphery during MI stages. The FMNL2 antibody staining results also confirmed this localization pattern. In addition, we co-stained FMNL2 antibody with F-actin, and the results revealed that both FMNL2 and F-actin are localized in the cortex region of oocytes (*Figure 1C*). Similar localization was also found in porcine oocytes (*Figure 1—figure supplement 1*). The FMNL2 localization pattern indicated that FMNL2 might interact with actin dynamics during oocyte meiosis.

### FMNL2 is essential for polar body extrusion and asymmetric division in oocytes

To investigate the functional roles of FMNL2 in mouse oocytes, we employed *Fmnl2* siRNA microinjection to knockdown FMNL2 protein expression. A significant decrease of FMNL2 protein level was

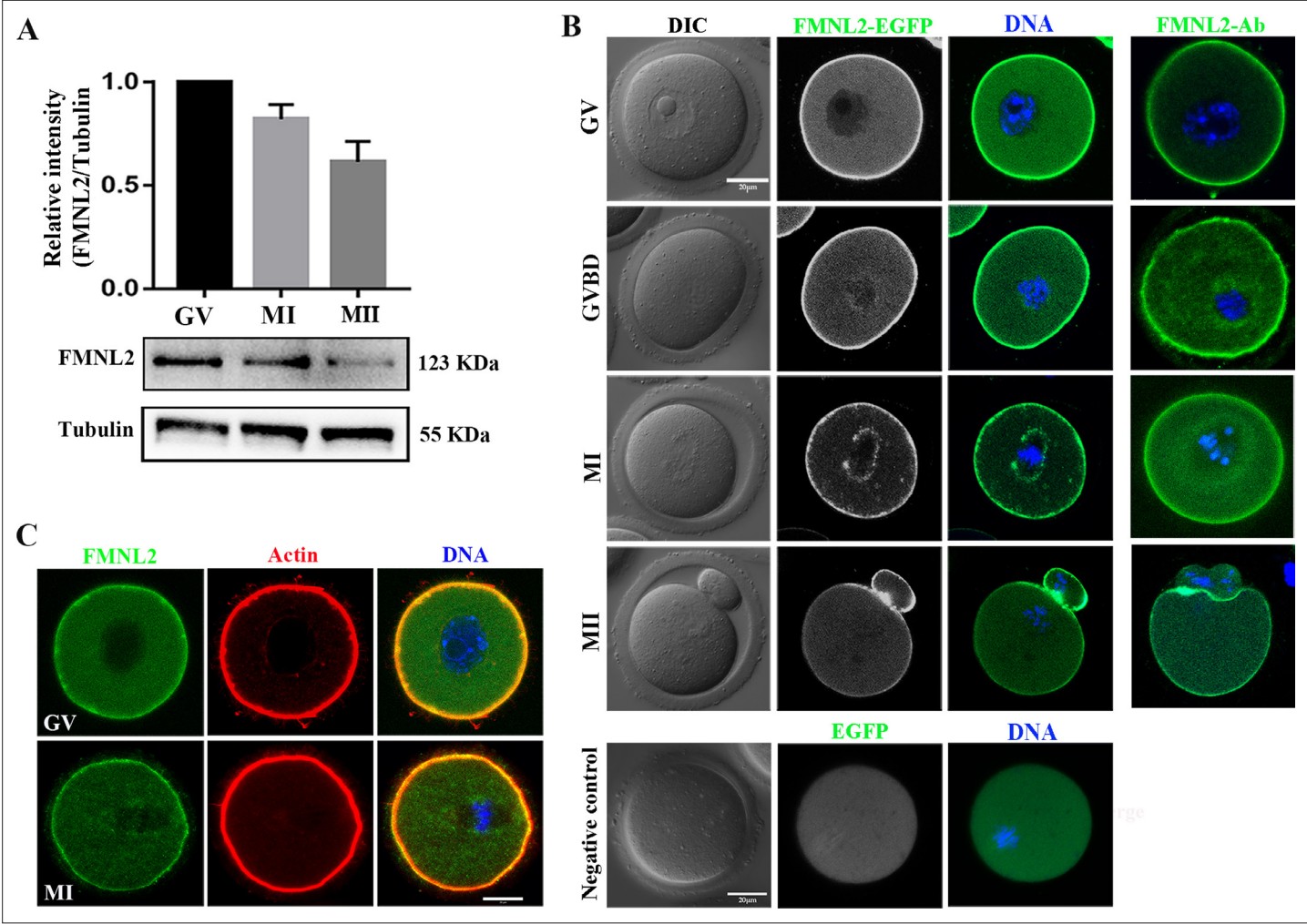

**Figure 1.** Expression and subcellular localization of FMNL2 during mouse oocyte meiosis. (**A**) Western blotting results of FMNL2 protein expression at different stages. FMNL2 expressed at the germinal vesicle (GV), metaphase I (MI), and metaphase II (MII) stages. (**B**) Subcellular localization of FMNL2-EGFP and FMNL2 antibody during mouse oocyte meiosis. FMNL2 was enriched at the cortex (GV, germinal vesicle breakdown [GVBD], MI, and MII stage) and spindle periphery (MI stage). Green, FMNL2-EGFP; blue, DNA. Negative control: green, EGFP; blue, DNA. Bar = 20 μm. (**C**) Co-staining of oocytes for FMNL2 and actin. FMNL2 and actin both localization in cortex. Green, FMNL2-antibody; red, actin; blue, DNA. Bar = 20 μm.

The online version of this article includes the following source data and figure supplement(s) for figure 1:

**Source data 1.** The original files of the full raw unedited blots in *Figure 1*.

**Source data 2.** The figure with the uncropped blots with the labeled bands.

**Figure supplement 1.** Localization of FMNL2 in the different stages of porcine oocyte maturation.

shown in FMNL2-KD oocytes compared to control group by western blot (1 vs. 0.48 ± 0.08, p < 0.01, *Figure 2A*). We then cultured the oocytes in vitro for 12 hr and examined the maturation of oocytes, and the results indicated that knockdown of FMNL2 had an impact on the extrusion of the first polar body. Moreover, a significant proportion of oocytes exhibited larger polar bodies upon extrusion (*Figure 2B*). Based on the size of the extruded polar bodies, those with a diameter exceeding one-third of the oocyte's diameter were categorized as large polar bodies. Consequently, we proceeded to calculate the rates of polar body extrusion and the generation of large polar bodies in the oocytes. The quantitative results also confirmed this phenotype (rate of polar body extrusion: 74.26 ± 1.44%, *n* = 439 vs. 59.5 ± 2.82%, *n* = 398, p < 0.001, *Figure 2C*; rate of large polar bodies: 19.05 ± 1.97%, *n* = 311 vs. 37.16 ± 1.87%, *n* = 257, p < 0.0001, *Figure 2D*). In addition, live-cell imaging was used to determine the dynamic changes that occurred during oocyte maturation, and the results showed that the oocytes either failed to undergo cytokinesis or divided from the central axis of the oocytes

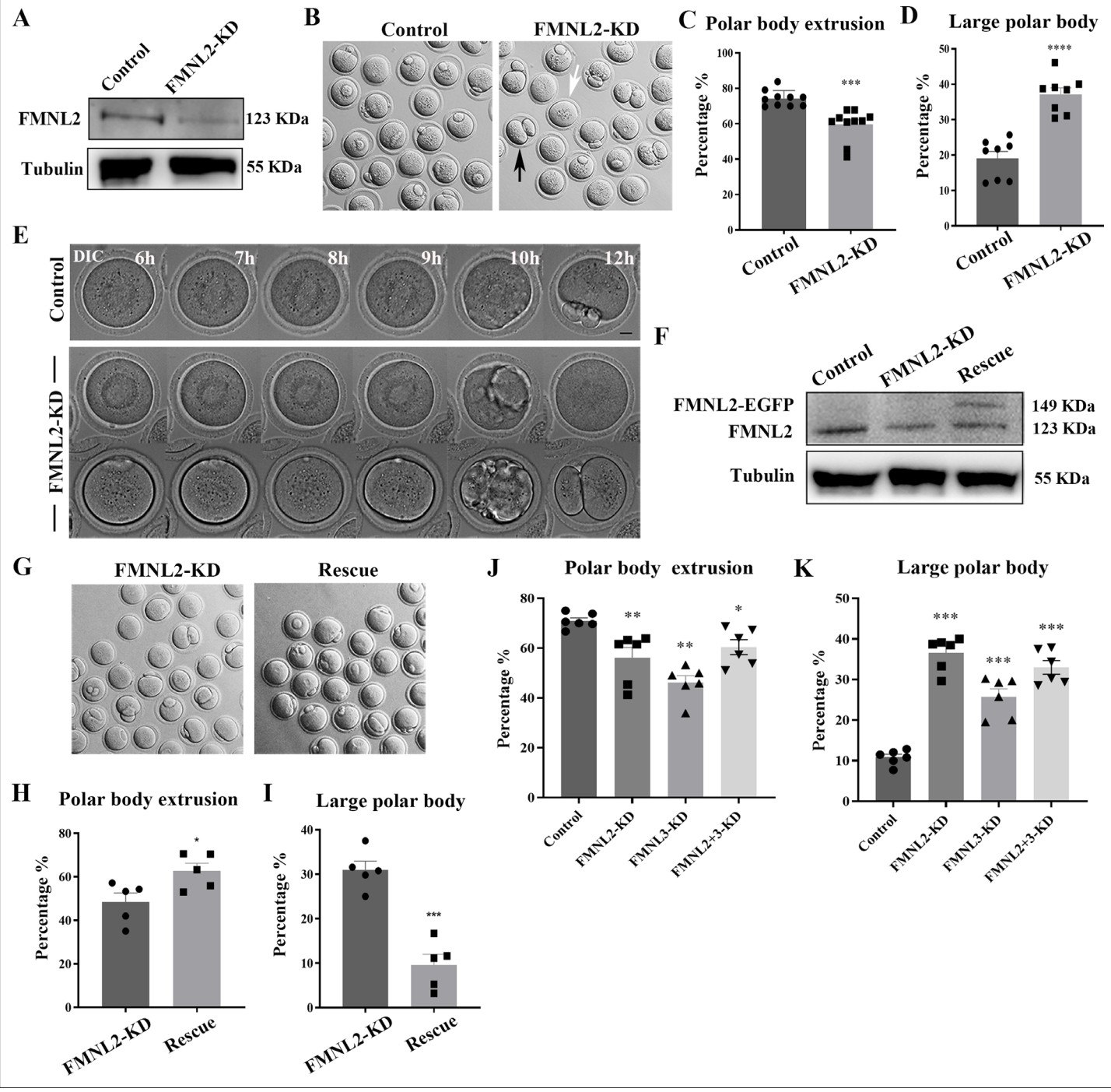

**Figure 2.** Knockdown of FMNL2 affects first polar body extrusion and asymmetric division. (**A**) Western blot analysis for FMNL2 expression in the FMNL2-KD group and control group. Relative intensity of FMNL2 and tubulin was assessed by densitometry. (**B**) Brightfield images of control oocytes and FMNL2-KD oocytes after 12 hr culture. FMNL2-KD caused large polar bodies (black arrows) and some oocytes failed to extrude the polar bodies (white arrows). (**C**) Rate of polar body extrusion after 12 hr culture of the control group and FMNL2-KD group. Control (n = 439), FMNL2-KD (n = 398) . (**D**) Rate of large polar body extrusion after 12 hr culture in the control group and FMNL2-KD group. Control (n = 311), FMNL2-KD (n = 257). (**E**) Time-lapse microscopy showed that polar body extrusion failed after FMNL2-KD. Bar = 10 µm. (**F**) Western blot analysis for FMNL2 expression in the control group, FMNL2-KD group, and rescue group. Relative intensity of FMNL2 and tubulin was assessed by densitometry. (**G**) Brightfield images of FMNL2-KD oocytes and rescue oocytes after 12 hr culture. (**H**) Rate of polar body extrusion after 12 hr culture of the FMNL2-KD group and rescue group.FMNL2-KD (n = 355), Rescue (n = 377). (**I**) Rate of large polar body extrusion after 12 hr culture in the FMNL2-KD group and rescue group. FMNL2-KD (n = 193), Rescue (n = 203). (**J**) Rate of polar body extrusion after 12 hr culture of the control group, FMNL2-KD group，FMNL3-KD group and FMNL2 + 3-KD group. Control (n = 261), FMNL2-KD (n = 203 ), FMNL3-KD (n = 184), FMNL2+3-KD (n = 198). (**K**) Rate of large polar body extrusion after 12 hr culture

*Figure 2 continued on next page*

*Figure 2 continued*

in the control group, FMNL2-KD group，FMNL3-KD group and FMNL2 + 3-KD group. Control (n = 172), FMNL2-KD (n = 178), FMNL3-KD (n = 136), FMNL2+3-KD (n = 118). The error bars are representing the mean ± SEM. The P-values were calculated using Student's t-test. *p < 0.05, **p < 0.01, ***p < 0.001, ****p < 0.0001.

The online version of this article includes the following source data for figure 2:

**Source data 1.** The original files of the full raw unedited blots in *Figure 2*.

**Source data 2.** The figure with the uncropped blots with the labeled bands.

and formed big polar bodies (*Figure 2E*). To further confirm the phenotype, we performed FMNL2 rescue experiments by expressing exogenous *Fmnl2* mRNA in FMNL2-depleted oocytes (*Figure 2F*), we found that exogenous *Fmnl2* mRNA expression rescued first polar body extrusion and large polar body defects (*Figure 2G*). The quantitative results also confirmed this phenotype (rate of polar body extrusion: 48.34 ± 4.2%, *n* = 355 vs. 62.62 ± 3.6%, *n* = 377, p < 0.01, *Figure 2H*; rate of large polar bodies: 30.93 ± 2%, *n* = 193 vs. 9.58 ± 2.4%, *n* = 203, p < 0.01, *Figure 2I*). It is known that knockdown of FMNL3 leads to inhibition of oocyte maturation. To investigate whether FMNL2 exhibits an additive effect with FMNL3 in terms of functionality, we simultaneously knocked down both FMNL2 and FMNL3. The results demonstrated that simultaneous knockdown of the two FMNL proteins, compared to the control group, resulted in a decrease in oocyte maturation rate. However, when compared to the single knockdown of *Fmnl2*, the double knockdown of FMNL2 and FMNL3 did not cause more severe defects in polar body extrusion (polar body extrusion, Control: 70.97 ± 1.23%, *n* = 261 vs. FMNL2 + 3-KD: 60.42 ± 2.99%, *n* = 198, p < 0.05, *Figure 2J*; large polar body, Control: 10.85 ± 0.97%, *n* = 172 vs. FMNL2 + 3-KD: 32.90 ± 1.88%, *n* = 118, p < 0.001, *Figure 2K*). These results suggested that FMNL2 played critical roles for the polar body extrusion and asymmetric division during mouse oocyte maturation.

## FMNL2 regulates meiotic spindle migration during oocyte maturation

To investigate the causes for polar body extrusion defects, we examined the spindle migration during oocyte meiosis using time-lapse microscopy after culturing oocytes in vitro for 8 hr. As shown in *Figure 3A*, in the control oocyte, the meiotic spindle formed in the center of the oocyte after culture 8 hr and moved to the oocyte cortex at 9.5 hr; and the polar body was extruded at 11–12 hr, with a spindle formed near the cortex at MII stage. However, in FMNL2-KD oocytes, two phenotypes were observed: (1) the meiotic spindle remained in the center of the oocyte until 10 hr, and then the oocytes initiated the cytokinesis at 10.5 hr but failed to extrude the polar body; (2) some oocytes with arrested spindles initiated the cytokinesis but extruded a big polar body (*Figure 3A*). This indicated the failure of spindle migration after FMNL2 depletion. We analyzed the rate of cortex-localized spindle in oocytes by cultured for 9.5 hr, and the result showed that the rate of migrated spindles in control oocytes was significantly higher than that in FMNL2-KD oocytes (59.94 ± 3.42%, *n* = 78 vs. 38.97 ± 6.34%, *n* = 64, p < 0.05, *Figure 3B*). We also performed FMNL2 rescue experiments. Supplementing with exogenous *Fmnl2* rescued the spindle migration defects compared with the FMNL2-depletion group (40.27 ± 3.19%, *n* = 81 vs. 57.01 ± 2.72%, *n* = 57, p < 0.01, *Figure 3C*). We also quantified the extent of spindle migration: we regarded the oocyte diameter as *D* and the distance from the spindle pole to the oocyte cortex as *L*, with the *L/D* ratio indicating the extent of spindle migration to the cortex. Our results indicated that by cultured for 9.5 hr, the *L/D* ratio of the FMNL2-KD oocytes was significantly greater than that of the control oocytes (0.17 ± 0.05, *n* = 18 vs. 0.24 ± 0.05, *n* = 18, p < 0.001; *Figure 3D*). Supplementing with exogenous FMNL2 rescued the spindle migration defects compared with the FMNL2-depletion group (0.26 ± 0.04, *n* = 12 vs. 0.16 ± 0.04, *n* = 13, p < 0.0001; *Figure 3D*). Similar results were also observed in porcine oocytes (*Figure 3—figure supplement 1*). These results suggested that FMNL2 might be involved in spindle migration in mouse and porcine oocytes.

## FMNL2 promotes cytoplasmic actin assembly during oocyte maturation

As FMNL2 is a key actin assembly factor, we further investigated actin assembly after deleting FMNL2 in mouse oocytes. After culturing oocytes in vitro for 9 hr, the MI oocytes were stained with Phalloidin-TRITC. Surprisingly, there was no significant difference for the signals of cortex actin was observed

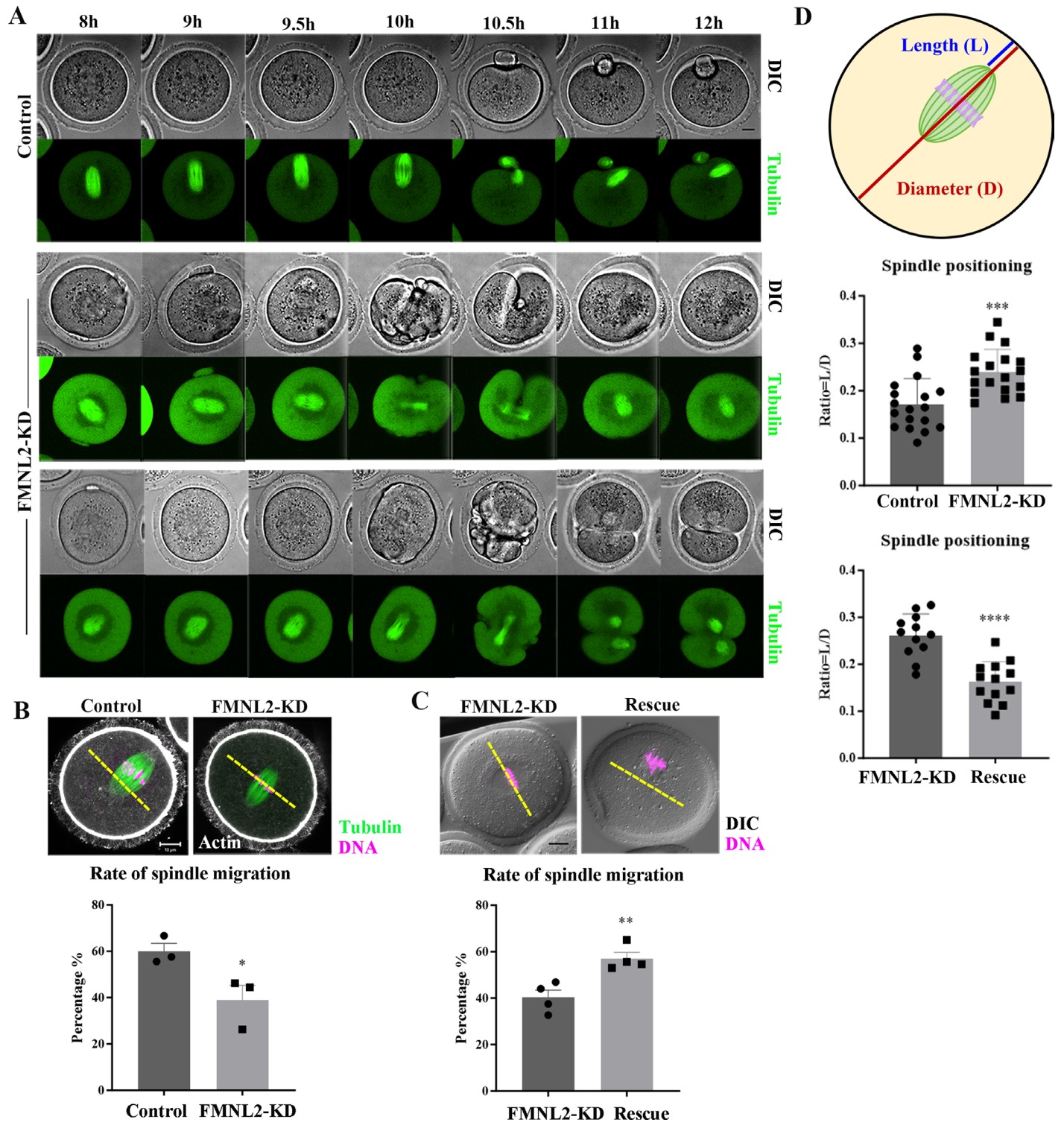

**Figure 3.** Knockdown of FMNL2 disrupts spindle localization during mouse oocyte meiosis. (**A**) Time-lapse microscopy showed that spindle migration failed after FMNL2-KD. Green, tubulin-EGFP. Bar = 10 μm. (**B**) Representative images and the proportion of spindle migration after 9.5 hr of culture in the control group and FMNL2-KD oocyte group. White, actin; green, tubulin; magenta, DNA. Bar = 10 μm. Control (n = 78), FMNL2-KD (n = 64 ). (**C**) Representative images and the proportion of spindle migration after 9.5 hr of culture in the FMNL2-KD group and rescue oocyte group. Magenta, DNA. Bar = 10 μm. FMNL2-KD (n = 81), Rescue (n = 57). (**D**) Quantitative analysis of the extent of spindle migration. Control (n = 18), FMNL2-KD (n = 18); FMNL2-KD (n = 12), Rescue (n = 13). The error bars are representing the mean ± SEM. The P-values were calculated using Student's t-test. *p < 0.05, **p < 0.01. ***p < 0.001, ****p < 0.0001.

*Figure 3 continued on next page*

*Figure 3 continued*

The online version of this article includes the following figure supplement(s) for figure 3:

**Figure supplement 1.** The spindle positioning after FMNL2 antibody injection in porcine oocytes.

between control oocytes and FMNL2-KD oocytes, which was confirmed by the fluorescence intensity analysis (30.88 ± 1.10, *n* = 28 vs. 30.58 ± 1.12, *n* = 28, p > 0.05, *Figure 4A, B*). However, we found a significant decrease of cytoplasmic actin signals in the FMNL2-KD oocytes, and the statistical analysis for the cytoplasmic actin fluorescent signals also confirmed our findings (58.25 ± 2.05, *n* = 26 vs. 37.92 ± 2.02, *n* = 24, p < 0.0001, *Figure 4C, D*). Similar results were also observed in porcine oocytes (*Figure 4—figure supplement 1*). Moreover, the rescue experiments showed that exogenous FMNL2 rescued the decrease of cytoplasmic actin filaments compared with the FMNL2-depletion group (37.98 ± 1.98, *n* = 16 vs. 54.72 ± 2.88, *n* = 15, p < 0.0001, *Figure 4E, F*). We next explored how FMNL2 regulates cytoplasmic actin assembly in oocytes. By mass spectrometry analysis, we found there were several actin-related potential candidates which might be related with FMNL2 (*Figure 4G*). Co-immunoprecipitation results showed that FMNL2 precipitated Arp2 and Formin2 but not Profilin and fascin (*Figure 4H*). To further verify the correlation between FMNL2 and Arp2 and Formin2, we then examined Arp2 and Formin2 protein expression after FMNL2 knockdown. The results showed Arp2 protein expression increased significantly after FMNL2 knockdown (1 vs. 1.56 ± 0.07, p < 0.001, *Figure 4I*) but Formin2 decreased after FMNL2 knockdown (1 vs. 0.62 ± 0.04, p < 0.001, *Figure 4J*). Exogenous FMNL2 rescued these alterations compared with that in the FMNL2-KD group (Arp2 protein expression: 1 vs. 0.65 ± 0.06, p < 0.01, *Figure 4I*; Formin2 protein expression: 1 vs. 1.24 ± 0.05, p < 0.01, *Figure 4J*). These results indicated that FMNL2 may be associated with Formin2 and Arp2 for actin assembly in mouse and porcine oocytes.

## FMNL2 regulates ER distribution during oocyte maturation

Ovarian mass spectrometry analysis data indicated that several ER-related potential candidates which might be related with FMNL2 (*Figure 5A*), while INF2, a typical protein which mediates actin polymerization at ER showed high confidence level. Therefore, we speculated if there is consistency in oocytes as in the ovary, and thus we examined the relationship between FMNL2 and INF2 in oocytes. The co-immunoprecipitation results showed that FMNL2 precipitated INF2 and INF2 also precipitated FMNL2 (*Figure 5B*), indicating that FMNL2 interacted with INF2 in mouse oocytes. We then examined the ER distribution after culturing oocytes in vitro for 9 hr. As shown in *Figure 5C*, in control oocytes the ER is highly concentrated around the spindle; however, in the FMNL2-KD oocytes, in addition to being concentrated around the spindle, the ER also forms clusters in the cytoplasm (*Figure 5C*). We defined this phenomenon of having a large number of clustered ER in the cytoplasm as abnormal distribution. The statistical analysis showed that the abnormal distribution of ER increased significantly in the FMNL2-KD group (28.91 ± 5.62%, *n* = 27 vs. 59.64 ± 6.95%, *n* = 28, p < 0.05, *Figure 5D*). Similar results were also observed in porcine oocytes (*Figure 5—figure supplement 1*). The localization pattern of ER indicated its functions might be disturbed. In FMNL2-KD oocytes, we found the expressions of ER-stress-related proteins Grp78 and Chop were significantly increased (Grp78: 1 vs. 1.42 ± 0.12, p < 0.05; Chop: 1 vs. 1.53 ± 0.16, p < 0.05, *Figure 5E*), indicating the occurrence of ER stress. We also performed FMNL2 rescue experiments. Supplementing with exogenous *Fmnl2* mRNA rescued the ER distribution defects caused by FMNL2 knockdown (*Figure 5F*), which was supported by the statistical analysis showing that the abnormal distribution rate of ER decreased significantly in the rescue group (52.04 ± 5.29%, *n* = 70 vs. 34.91 ± 3.37%, *n* = 78, p < 0.05, *Figure 5G*). Moreover, Grp78 protein expression decreased in the rescue group (1 vs. 0.78 ± 0.05, p < 0.01, *Figure 5H*). These results indicated that the depletion of FMNL2 affected ER distribution and caused ER stress in mouse oocytes.

## FMNL2 regulates mitochondrial distribution during mouse oocyte maturation

As INF2 is also related to the mitochondrial connection of ER, we further screened up the mass spectrometry analysis data and we found many mitochondria-related potential candidates which might be related with FMNL2 (*Figure 6A*). Therefore, we further examined the distribution of mitochondria

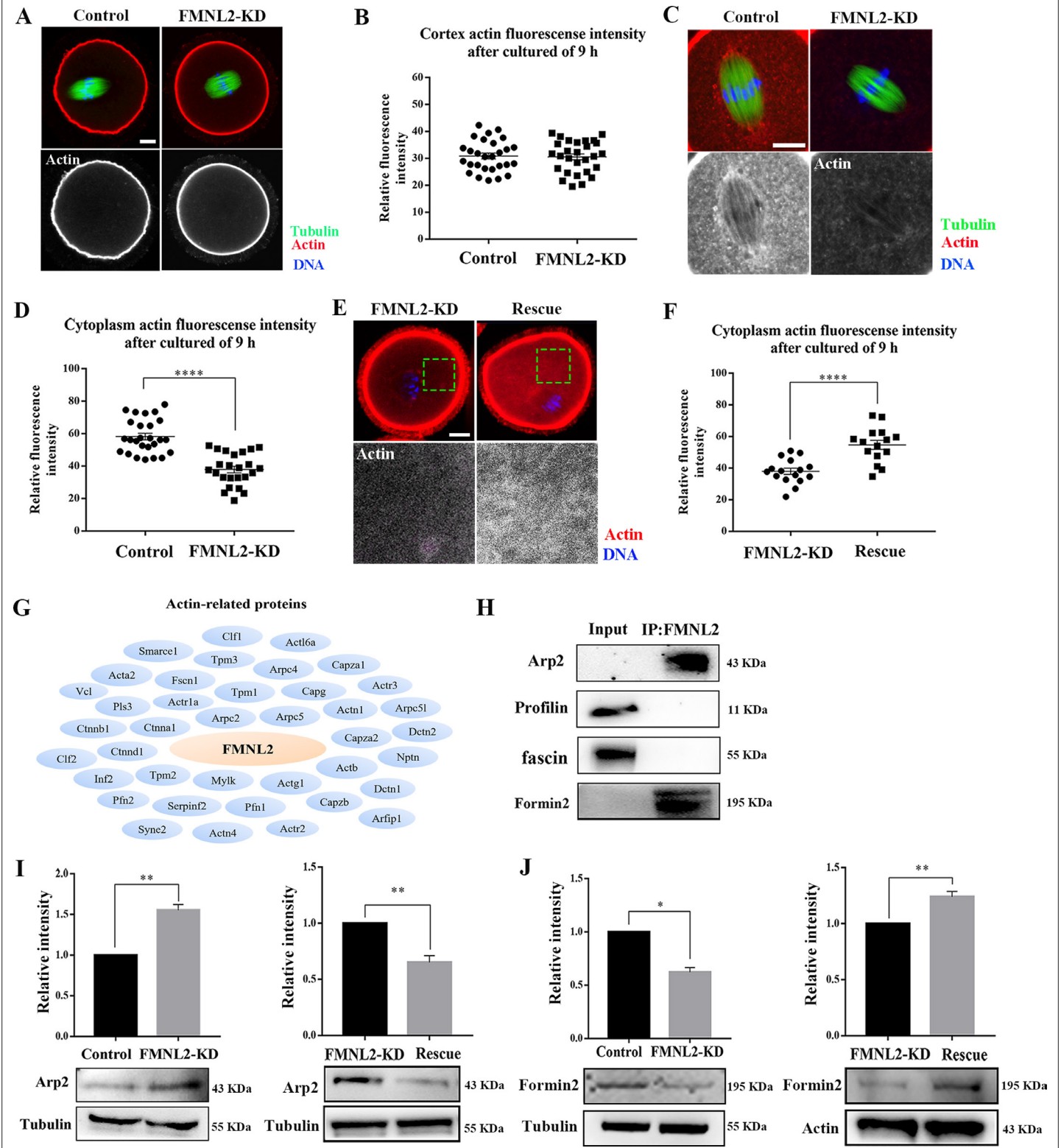

**Figure 4.** Knockdown of FMNL2 disrupts actin assembly during mouse oocyte meiosis. (**A, B**) Representative images of actin distribution at the oocyte cortex and the fluorescent intensities in the control group and FMNL2-KD group (p > 0.1). White, actin; green, tubulin; blue, DNA. Bar = 10 μm. Control (n = 28), FMNL2-KD (n = 28). (**C, D**) Representative images of actin distribution in the oocyte cytoplasm and the fluorescent intensities in the control group and FMNL2-KD group. White, actin; green, tubulin; blue, DNA. Bar = 10 μm. Control (n = 26), FMNL2-KD (n = 24). (**E, F**) Representative images of actin distribution in the oocyte cytoplasm and the fluorescent intensities in the FMNL2-KD group and rescue group. White, actin; blue, DNA. Bar =

*Figure 4 continued on next page*

*Figure 4 continued*

10 µm. FMNL2-KD (n = 16), Rescue (n = 15). (**G**) Mass spectrometry results showed that FMNL2 was related to many actin-related proteins. (**H**) Co-IP results showed that FMNL2 was correlated with Arp and Formin2 but not with Profiling and Fascin. (**I**) Arp2 protein expression significantly increased in the FMNL2-KD oocytes compared with the control oocytes. Arp2 protein expression significantly decreased in the rescue oocytes compared with the FMNL2-KD oocytes. (**J**) Formin2 protein expression significantly decreased in the FMNL2-KD oocytes compared with the control oocytes. Formin2 protein expression significantly increased in the rescue oocytes compared with the FMNL2-KD oocytes. The error bars are representing the mean ± SEM. The P-values were calculated using Student's t-test. *p < 0.05, **p < 0.01, ****p < 0.0001.

The online version of this article includes the following source data and figure supplement(s) for figure 4:

**Source data 1.** The original files of the full raw unedited blots in *Figure 4*.

**Source data 2.** The figure with the uncropped blots with the labeled bands.

**Source data 3.** The original file of mass spectrometry for the protein summary.

**Figure supplement 1.** The actin intensity in the cytoplasm of porcine oocytes.

after culturing oocytes in vitro for 9 hr. In control oocytes, the mitochondria evenly distributed in the cytoplasm and accumulated at the spindle periphery in MI stage; however, in FMNL2-KD oocytes, in addition to being concentrated around the spindle, the mitochondria also forms clusters in the cytoplasm (*Figure 6B*). We counted the number of clumps in cytoplasm and found that the uniform distribution of mitochondria decreased significantly in the FMNL2-KD group (59.66 ± 8.48%, *n* = 31 vs. 20.83 ± 4.17%, *n* = 32, p < 0.05, *Figure 6C*). A large number of FMNL2-KD oocytes agglomerated into one to three clumps (22.73 ± 4.27%, *n* = 31 vs. 42.50 ± 1.25%, *n* = 32, p < 0.05, *Figure 6C*). Similar results were also observed in porcine oocytes (*Figure 6—figure supplement 1*). Supplementing with exogenous *Fmnl2* mRNA rescued the mitochondria distribution (*Figure 6D*), the statistical analysis showed that the uniform distribution of mitochondria increased significantly in the rescue group (36.49 ± 3.97%, *n* = 53 vs. 53.90 ± 2.09%, *n* = 79, p < 0.05, *Figure 6E*). We also examined mitochondrial membrane potential (MMP), and the results showed that FMNL2 depletion caused the alterations of MMP by JC-1 staining after culturing oocytes in vitro for 9 hr. The fluorescence intensity of JC-1 red channel was decreased compared with the control group (*Figure 6F*). We also calculated the ratio for red/green fluorescence intensity, and the results also confirmed this (control group: 0.40 vs. FMNL2-KD: 0.21 ± 0.01, p < 0.01) (*Figure 6G*). Cofilin is an important factor of actin assembly and regulates mitochondrial function. We also examined cofilin protein expression after FMNL2 knockdown. The results showed cofilin protein expression decreased significantly after FMNL2 knockdown (1 vs. 0.81 ± 0.03, p < 0.01, *Figure 6H*). These results indicated that FMNL2 regulated mitochondria distribution and function during mouse and porcine oocyte maturation.

## Discussion

In this study, we explored the functions of FMNL2 during mouse and porcine oocyte meiosis. Our results indicated that FMNL2 regulated actin-based spindle migration for asymmetric cell division of oocytes, and more importantly FMNL2 was critical for maintaining the distribution of the ER and mitochondria, which set up a link for actin-related spindle migration and organelle dynamics in mammalian oocytes (*Figure 7*).

As a subfamily of Formin family, FMNLs play an important role in regulating actin filaments (*Breitsprecher and Goode, 2013*), while FMNL2 is most widely expressed in variety of cell models among the members of FMNLs. In this study, we showed that FMNL2 expressed in mouse oocytes and it mainly accumulated at the oocyte cortex and spindle periphery, which was similar with the actin distribution pattern in oocytes. This specific localization is also similar to FMN2, a well-studied factor in the formin family for spindle migration during oocyte meiosis (*Duan et al., 2020*; *Li et al., 2008*). In addition, another FMNLs family member, FMNL1 is also localized at the cortex and is essential for actin polymerization and spindle assembly during oocyte meiosis (*Wang et al., 2015*). Based on the localization pattern of FMNL2, we speculated that the functions of FMNL2 might be also involved in actin-related process during mouse oocyte meiosis.

To confirm our hypothesis, we depleted FMNL2 protein expression and we found that absence of FMNL2 caused the aberrant first polar body extrusion. The oocytes either failed to form the polar body or extruded large polar bodies. These phenotypes caused by FMNL2 depletion are similar

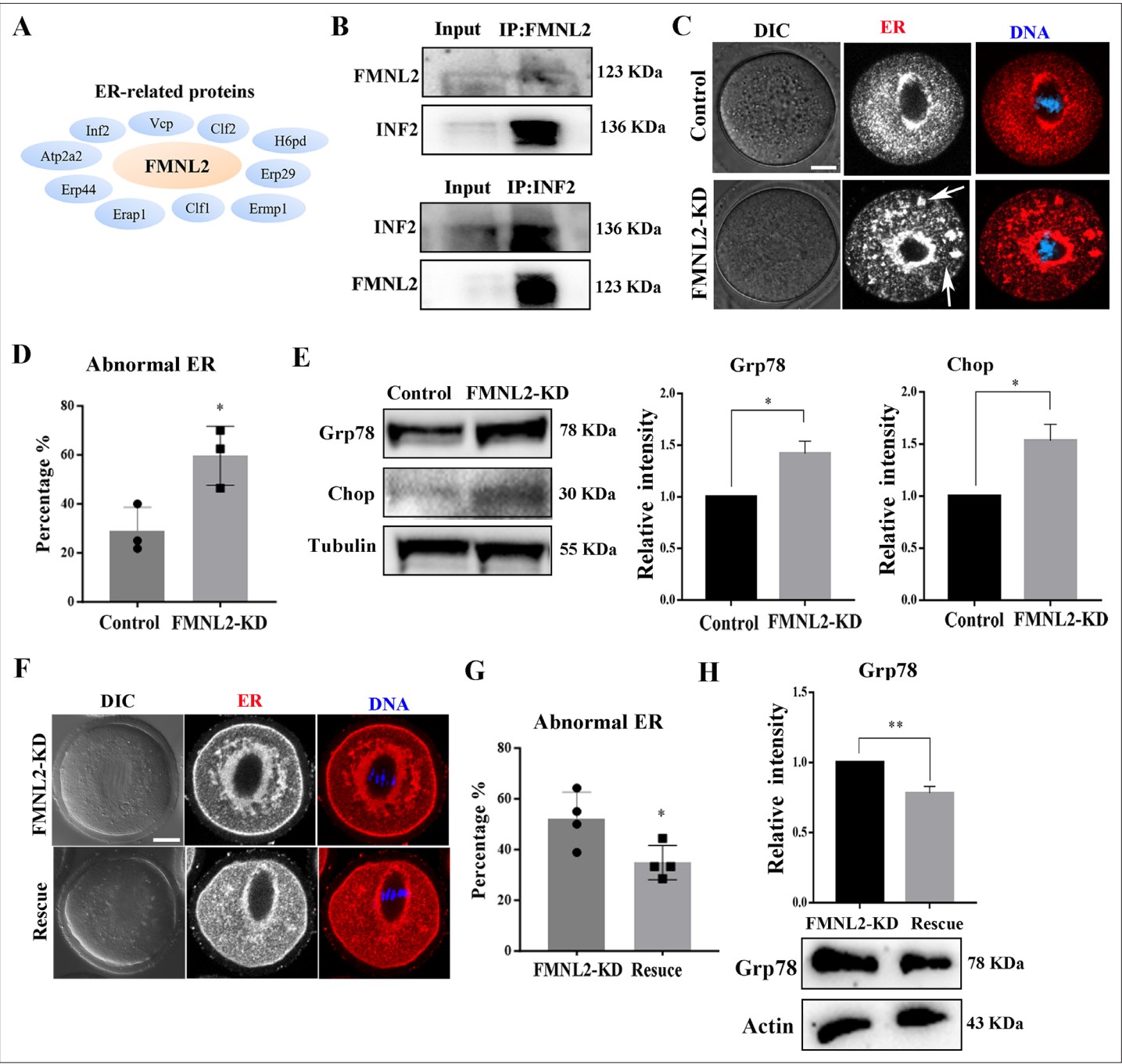

**Figure 5.** FMNL2 regulates endoplasmic reticulum (ER) distribution during mouse oocytes maturation. (**A**) Mass spectrometry results showed that FMNL2 was associated with ER-related proteins. (**B**) Co-IP results showed that FMNL2 was correlated with INF2. (**C**) Representative images of ER distribution in the oocyte cytoplasm in the control group and FMNL2-KD group. In FMNL2-KD oocytes, ER agglomerated in cytoplasm (white arrow). Red, ER; blue, DNA. Bar = 20 μm. (**D**) Abnormal distribution of ER significantly increased in the FMNL2-KD oocytes compared with the control oocytes. Control (n = 27), FMNL2-KD (n = 28). (**E**) Grp78 and Chop protein expression significantly increased in the FMNL2-KD oocytes compared with the control oocytes. The band intensity analysis also confirmed this finding. (**F**) Representative images of ER distribution in the oocyte cytoplasm in the FMNL2-KD group and rescue group. Red, ER; blue, DNA. Bar = 20 μm. (**G**) Abnormal distribution of ER significantly decreased in the rescue oocytes compared with the FMNL2-KD oocytes. FMNL2-KD (n = 70), Rescue (n = 78). (**H**) Grp78 protein expression significantly decreased in the rescue oocytes compared with the FMNL2-KD oocytes. The error bars are representing the mean ± SEM. The P-values were calculated using Student's t-test. *p < 0.05, **p < 0.01.

The online version of this article includes the following source data and figure supplement(s) for figure 5:

**Source data 1.** The original files of the full raw unedited blots in *Figure 5*.

**Source data 2.** The figure with the uncropped blots with the labeled bands.

*Figure 5 continued on next page*

*Figure 5 continued*

**Figure supplement 1.** The endoplasmic reticulum (ER) distribution in porcine oocytes.

to the other actin-related proteins during oocyte maturation such as Arp2/3 complex (*Sun et al., 2011*; *Yi et al., 2011*) and FMN2 (*Dumont et al., 2007*; *Leader et al., 2002*). We next examined the actin distribution in oocytes since it is reported that FMNL2 promotes actin filament assembly in many models. FMNL2 is required for cell–cell adhesion formation by regulating the actin assembly (*Grikscheit et al., 2015*), and FMNL2 could directly drives actin elongation (*Block et al., 2012*). In CRC cells, cortactin bind to FMNL2 to active the actin polymerization, and FMNL2 is important for invadopodia formation and functions (*Ren et al., 2018*). Our results showed that the FMNL2 depletion caused significantly decrease in cytoplasmic actin, indicating the conserved roles of FMNL2 on actin assembly in mammalian oocyte model. Other Formin family proteins such as Daam1, FHOD1, and Formin-homology family protein mDia1 are also reported to affect oocyte meiosis by regulating actin polymerization (*Lu et al., 2017*; *Pan et al., 2018*; *Zhang et al., 2015*).

We then tried to explore how FMNL2 involves into the actin assembly in oocytes. Mass spectrometry analysis data indicated that FMNL2 associated with several actin-related proteins, and we found that a potential association between FMNL2 and Arp2/Formin2. This could be confirmed by the altered expression of these two molecules after FMNL2 depletion. Therefore, we speculated FMNL2 could regulate cytoplasmic actin assembly in oocytes through the association with Formin2 since it is reported to be an important protein for cytoplasmic actin assembly in oocytes (*Dumont et al., 2007*). Interestingly, our results showed that unlike the reduction of cytoplasmic actin, cortex actin was not affected by the absence of FMNL2. We speculate that FMNL2 and Arp2/3 both contribute to the cortex actin dynamics, when FMNL2 decreases, ARP2 increases to compensate for this, which maintains the cortex actin level. As an actin nucleator Arp2/3 complex localizes at the cortex and is essential for actin polymerization during oocyte meiosis (*Goley and Welch, 2006*; *Sun et al., 2011*). These results suggested that FMNL2 might be involved in cytokinesis and asymmetric division by regulating actin assembly during mouse oocyte maturation.

The spindle migration is a key step in ensuring the asymmetric division for oocytes (*Brunet and Maro, 2005*). In mitosis, spindle position is decided by cortical actin and astral microtubules; in contrast, spindle migration is mainly mediated by actin filaments during oocyte meiosis (*Brunet and Maro, 2005*; *Reinsch and Gönczy, 1998*). Due to the effects of FMNL2 on asymmetric division and cytoplasmic actin, we analyzed the spindle positioning at late MI, we found that the spindle migration was disturbed after FMNL2 depletion, no matter the cytokinesis occurred or not. Several formin proteins are shown to regulate spindle migration during oocytes meiosis. For example, FMN2 nucleates actin surrounding the spindle, pushing force generated by actin to trigger the spindle migration (*Duan et al., 2020*; *Dumont et al., 2007*), and cyclin-dependent kinase 1 (Cdk1) induces cytoplasmic Formin-mediated F-actin polymerization to propel the spindle into the cortex (*Wei et al., 2018*). Our previous studies also showed that absence of the formin family member FMNL1 or FHOD1 could lead to the decrease of cytoplasmic actin to prevent the spindle migration (*Pan et al., 2018*; *Wang et al., 2015*). We speculated that FMNL2 together with other Formin proteins, conservatively regulate actin-mediated spindle migration during oocyte meiosis.

Another important finding is that through the mass spectrometry analysis we found many candidate proteins which were related with ER, and our results indicated that FMNL2 was essential for the maintenance of ER distribution in the cytoplasm. Moreover, the loss of FMNL2-induced ER stress, showing with altered expression of GRP78 and CHOP. Proper distribution of ER is important for the oocyte quality. ER displays a homogeneous distribution pattern throughout the entire ooplasm during development of oocytes and embryos from diabetic mice (*Zhang et al., 2013*). During the transition of mouse oocytes from MI to MII phase, actin regulates cortical ER aggregation (*FitzHarris et al., 2007*). In addition, Formin2 is shown to colocalize with the ER during oocyte meiosis and the ER-associated Formin2 at the spindle periphery is required for MI chromosome migration (*Yi et al., 2013b*). In our results, we showed that FMNL2 associated with INF2 protein in oocytes. INF2 is an ER-associated protein, and the expression of GFP-INF2 which containing DAD/WH2 mutations causes the ER to collapse around the nucleus (*Chhabra et al., 2009*). We concluded that FMNL2 might regulate INF2 for the distribution of ER in cytoplasm of oocytes.

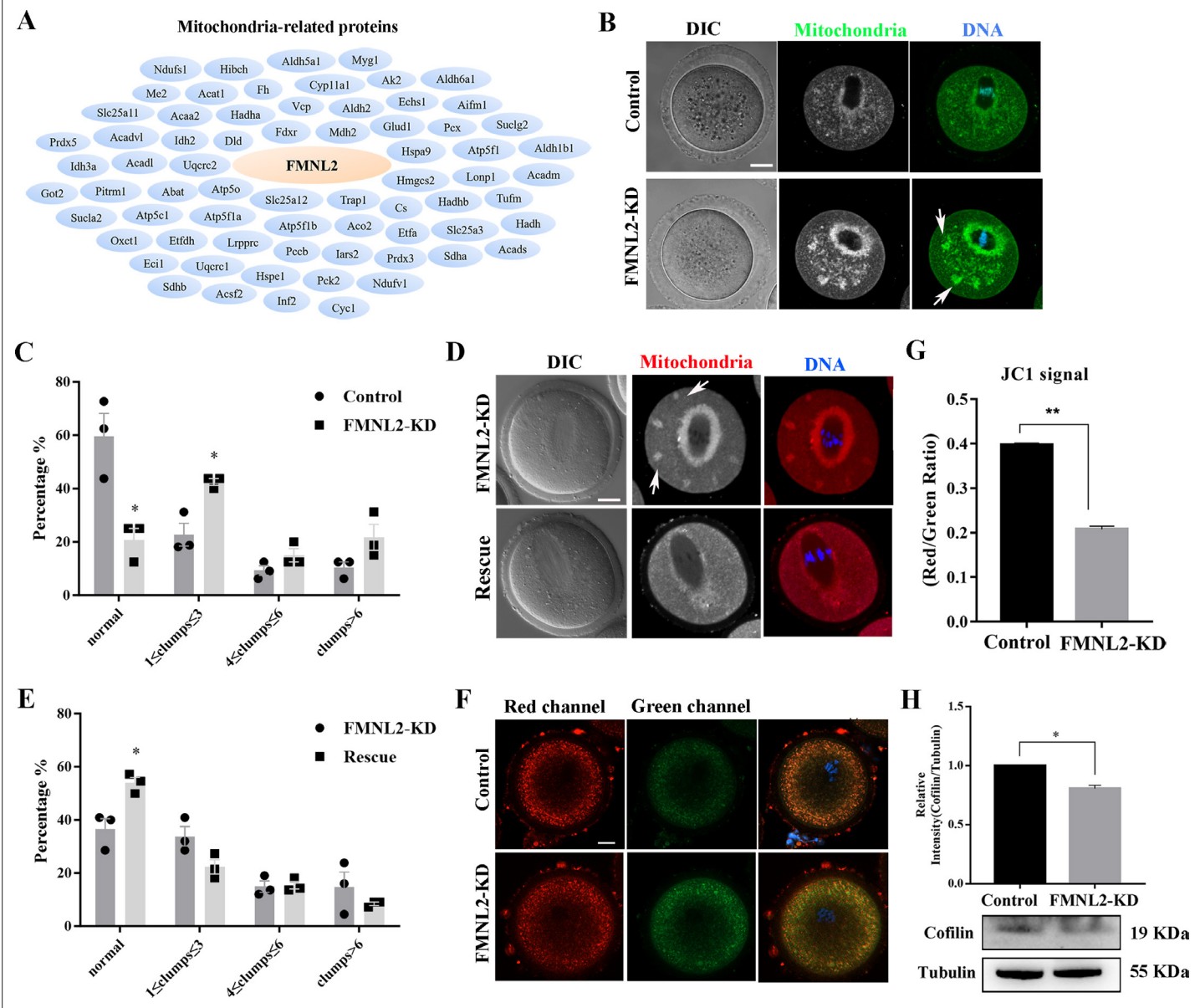

**Figure 6.** FMNL2 regulates mitochondrial distribution during mouse oocytes maturation. (**A**) Mass spectrometry results showed that FMNL2 was related to many mitochondria-related proteins. (**B**) Representative images of mitochondrial distribution in the oocyte cytoplasm in the control group and FMNL2-KD group. In FMNL2-KD oocytes, mitochondrial agglomerated in cytoplasm (white arrow). Green, Mito；blue, DNA. Bar = 20 µm. (**C**) Abnormal distribution of mitochondrial significantly increased in the FMNL2-KD oocytes compared with the control oocytes. Control (n = 31), FMNL2-KD (n = 32). (**D**) Representative images of mitochondrial distribution in the oocyte cytoplasm in the FMNL2-KD group and rescue group. In FMNL2-KD oocytes, mitochondrial agglomerated in cytoplasm (white arrow). Red, Mito; blue, DNA. Bar = 20 µm. (**E**) Abnormal distribution of mitochondrial significantly decreased in the rescue oocytes compared with the FMNL2-KD oocytes. FMNL2-KD (n = 53), Rescue (n = 79). (**F**) The typical picture for JC1 green channel and red channel after FMNL2-KD. (**G**) The JC1 signal (red/green ratio) after FMNL2-KD compare with the control group, the JC-1 red/green fluorescence ratio was significantly reduced in FMNL2-KD groups. blue，DNA. Bar = 20 µm. (**H**) Cofilin protein expression significantly decreased in the FMNL2-KD oocytes compared with the control oocytes. The band intensity analysis also confirmed this finding. The error bars are representing the mean ± SEM. The P-values were calculated using Student's t-test. *p < 0.05, **p < 0.01.

The online version of this article includes the following source data and figure supplement(s) for figure 6:

**Source data 1.** The original files of the full raw unedited blots in *Figure 6*.

**Source data 2.** The figure with the uncropped blots with the labeled bands.

**Figure supplement 1.** The mitochondria distribution in porcine oocytes.

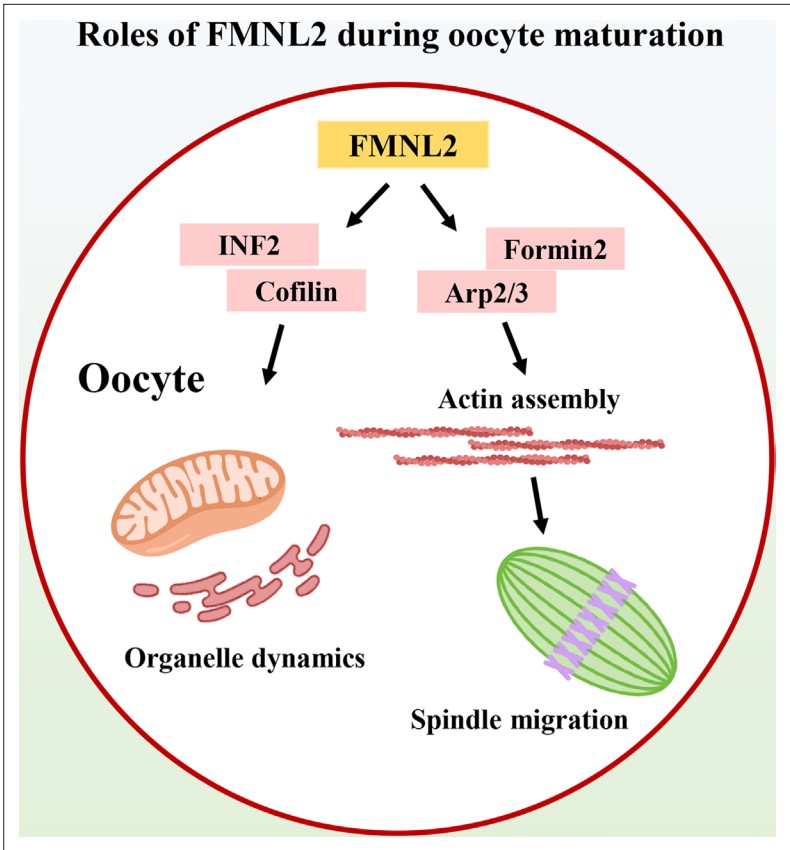

**Figure 7.** Diagram of the roles of FMNL2 during oocyte maturation. FMNL2 associates with Formin2 and Arp2/3 complex for actin assembly, which further regulates spindle migration and INF2/Cofilin-related organelle dynamics during mouse and porcine oocyte maturation.

Besides its roles of ER distribution, it is shown that INF2 also affects mitochondrial length and ER–mitochondrial interaction in an actin-dependent manner (*Chhabra et al., 2009*; *Korobova et al., 2013*). It is shown that INF2 regulates Drp1 for mitochondrial fission, and INF2-induced actin filaments may drive initial mitochondrial constriction, which allows Drp1-driven secondary constriction (*Ji et al., 2017*; *Korobova et al., 2013*). In addition, we also found many candidate proteins which were related with mitochondria from mass spectrometry analysis. During oocyte meiosis, mitochondria gradually accumulated around the spindle after GVBD, and the spindle-peripheral FMN2 and its actin nucleation activity are important for the accumulation of mitochondria in this region (*Duan et al., 2020*). Our results found that FMNL2 depletion caused agglutination of mitochondria and altered MMP level in the cytoplasm, indicating its roles on the mitochondria distribution and functions. Another formin protein mDia1 is shown to be necessary to induce the anchoring of mitochondria along the cytoskeletal in mammalian CV-1 cells and *Drosophila* BG2-C2 neuronal cells (*Minin et al., 2006*). Moreover, the formin interaction protein Spire1C binds INF2 to promote actin assembly on mitochondrial surfaces, and Spire1C disruption could reduce mitochondrial constriction and division (*Manor et al., 2015*). In addition, our result indicated that cofilin expression decreased in FMNL2-depletion oocytes. Cofilin is an actin-depolymerizing factor and its localization at the mitochondrial fission site is crucial for inducing mitochondrial fission and mitophagy (*Li et al., 2018*). Depleting of cofilin resulted in abnormal interconnection and elongation of mitochondria (*Li et al., 2015*). Together with its roles on ER, these data indicated that FMNL2 might associate with INF2 and cofilin for the actin-based organelle distribution during oocyte meiosis.

Collectively, we provide a body of evidence showing that FMNL2 associates with Formin2 and Arp2/3 complex for actin assembly, which further regulates spindle migration and INF2/Cofilin-related organelle dynamics during mammalian oocyte maturation.

# Materials and methods

## Antibodies and chemicals

Rabbit monoclonal anti-FMNL2 antibody, rabbit monoclonal anti-Arp2 antibody, mouse monoclonal anti-profilin1 antibody were from Santa Cruz (Santa Cruz, CA, USA). Rabbit monoclonal anti-Fascin antibody was purchased form Abcam (Cambridge, UK). Rabbit polyclonal anti-INF2 antibody was purchased from Proteintech (Proteintech, CHI, USA). Rabbit monoclonal anti-α-tubulin (11H10) antibody, rabbit monoclonal anti-Grp78 antibody, rabbit monoclonal anti-cofilin antibody, and rabbit monoclonal anti-Chop antibody were from Cell Signaling Technology (Beverly, MA, USA). Mouse monoclonal anti-α-tubulin-FITC antibody was from Sigma-Aldrich Corp (St. Louis, MO, USA). Fluorescein isothiocyanate (FITC)-conjugated goat anti-rabbit IgG were from Zhongshan Golden Bridge Biotechnology (Beijing). ER-Tracker Red, Mito-Tracker Green, and enhanced mitochondrial membrane potential assay Kit were from Beyotime Biotechnology (Shanghai). All other chemicals and reagents were from Sigma-Aldrich Corp, unless otherwise stated.

## Ethics statement and oocyte culture

We followed the guidelines of Animal Research Institute Committee of Nanjing Agricultural University to conduct the operations. The animal facility had license authorized by the experimental animal committee of Jiangsu Province (SYXK-Su-20170007). The Female Institute of Cancer Research (ICR) mice, aged 4–6 weeks, were kept in a room with a regulated temperature of 22°C and provided with a standard diet. Fully developed GV stage oocytes were retrieved from the ovaries of mice, and then cultured in M16 medium with paraffin oil at 37°C and in the presence of 5% $CO_2$ for in vitro maturation. At specific intervals, the oocytes were collected for various tests and analyses. The oocytes were placed at 37°C with an atmosphere of 5% $CO_2$, and cultured to different time points for immunostaining, microinjection, and western blot.

For porcine oocyte collection, the porcine ovaries were delivered from a local slaughterhouse in the thermos bottle within 2 hr. The cumulus cell complex (COCs) were acquired from 3 to 6 mm antral follicles, and cultured in TCM-199 medium for in vitro maturation from 4-well dish (Nunc, Denmark) at 38.5°C with an atmosphere of 5% $CO_2$. The porcine oocytes were collected at 24–28 hr for MI stage and 44–48 hr for MII stage.

## Plasmid construct and in vitro transcription

Template RNA was generated from mouse ovaries with RNA Isolation Kit (Thermo Fisher), then we reversed transcription of these RNA to create cDNA by a PrimeScript 1st strand cDNA synthesis kit (Takara, Japan). *Fmnl2-EGFP* vector was generated by Wuhan GeneCreate Biological Engineering Co, Ltd. mRNA was synthesized from linearized plasmid using HiScribe T7 high yield RNA synthesis kit (NEB), then capped with m7G (5′) ppp (5′) G (NEB) and tailed with a poly(A) polymerase tailing kit (Epicentre) and purified with RNA clean & concentrator-25 kit (Zymo Research).

## Microinjection of *Fmnl2* siRNA/mRNA and antibody

The *Fmnl2* siRNA working solution was dissolved in RNase-free water, achieving a concentration of 20 µM. For FMNL2 knockdown (KD), three individual siRNA strands were precisely mixed and subjected to centrifugation to obtain the supernatant, approximately 5–10 pl of supernatant was microinjected into the cytoplasm of GV-stage oocytes. In contrast, an equal volume of a negative control solution was microinjected into the cytoplasm of oocytes in the control group. *Fmnl2* siRNA: 5′-GCU GAA UGC UAU GAA CCU ATT-3′, 5′-GCC AUU GAU CUU UCU UCA ATT-3′, 5′-GGA AUU AAG AAG GCG ACA ATT-3′; Negative control siRNA: 5′-UUC UCC GAA CGU GUC ACG UTT-3′. Then, the oocytes were arrested in the GV stage for 18 hr in M16 medium with 5 µM milrinone. This was done to optimize the effectiveness of the siRNA and aid in the depletion of FMNL2. For the rescue experiment, 5–10 pl of 200 ng/µl *Fmnl2-EGFP* mRNA was injected into the GV oocytes 18 hr after *Fmnl2* siRNA injection. Following that, the GV-stage oocytes were cultured in M16 medium with 5 µM milrinone for 4 hr. Then, the oocytes being cultured in fresh M16 medium for subsequent experiments.

For FMNL2 antibody injection in the porcine oocytes, after injection the porcine oocytes were cultured in TCM-199 immediately. After 24–28 hr culture the oocytes were collected to stain actin, α-tubulin, ER, and mitochondria.

## Immunofluorescent staining and confocal microscopy

Oocytes were fixed in 4% paraformaldehyde (in phosphate-buffered saline [PBS]) for 30 min and permeabilized with 0.5% Triton X-100 in PBS for 20 min (1% Triton X-100 for overnight in porcine oocytes) then blocked in blocking buffer (1% bovine serum albumin-supplemented PBS) at room for 1 hr. For FMNL2 staining, the oocytes at various stages (GV, GVBD, MI, and MII) were incubated with Rabbit monoclonal anti-FMNL2 antibody (1:100) at 4°C overnight, then oocytes were washed by wash buffer (0.1% Tween 20 and 0.01% Triton X-100 in PBS) for three times (5 min each time). Next the oocytes were labeled with secondary antibody coupled to FITC-conjugated goat anti-rabbit IgG (1:100) at room temperature for 1 hr. For α-tubulin staining, MI stage oocytes were incubated with anti-α-tubulin-FITC antibody (1:200). For actin staining, GV and MI stage oocytes were incubated with Phalloidin-TRITC at room temperature for 2 hr. Then the oocytes were washed as the same way. Finally, oocytes were incubated with Hoechst 33342 at room temperature for 10–20 min. After staining, samples were mounted on glass slides and observed with a confocal laser-scanning microscope (Zeiss LSM 800 META, Germany).

## ER and Mito-tracker staining

To study ER and mitochondria distribution during mouse oocyte meiosis, MI stage oocytes were incubated with ER-Tracker Red (1:3000) or 200 nM Mito-tracker green (Red) in M16 medium (TCM-199 for porcine oocytes) for 30 min at 37°C and 5% $CO_2$. Then the oocytes were washed three times with M2 medium (TCM-199 for porcine oocytes), finally the samples were examined with confocal microscopy. During the MI stage of oocyte development, both the ER and mitochondria evenly distributed in the cytoplasm and accumulated at the spindle periphery in MI stage. The clustering pattern is considered as the abnormal localization pattern of these organelles.

## JC-1 detection

The enhanced mitochondrial membrane potential assay Kit was employed to analyze the MMP of oocytes. The MI stage oocytes were transferred from the M16 medium to JC-1 for 30 min at 37°C and 5% $CO_2$. Following three washes with M2 medium, the oocytes were examined using a fluorescent microscope (OLYMPUS IX71, Japan) for the presence of a fluorescent signal.

## Time-lapse microscopy

To image the dynamic changes that occurred during oocyte maturation, oocytes were cultured in M16 medium, then transferred to the Leica SD AF confocal imaging system equipped with 37°C incubator and 5% $CO_2$ supply (H301-K-FRAME). The spindle in oocytes was labeled by α-tubulin-EGFP.

## Immunoprecipitation

Four to six ovaries were put into Radio Immunoprecipitation Assay (RIPA) Lysis Buffer contained phosphatase inhibitor cocktail (100×) (Kangwei Biotechnology, China), and were completely cleaved on ice block. We collected supernatant after centrifugation (13,200 rps, 20 min) and then took out 50 μl as input sample at 4°C. The rest of the supernatant was incubated with primary antibody (FMNL2 or INF2 antibody) overnight at 4°C. 30 μl conjugated beads (washed five times in PBS) were added to the supernatant/antibody mixture and incubated at 4°C for 4–6 hr, after three times wash by immune complexes, the samples were then released from the beads by mixing in 2× sodium dodecyl sulfate (SDS) loading buffer for 10 min at 30°C.

## Western blot analysis

Approximate 100–150 GV or MI stage mouse oocytes were placed in Laemmli sample buffer and heated at 85°C for 7–10 min. Proteins were separated by SDS–polyacrylamide gel electrophoresis at 165 V for 70–80 min and then electrophoretically transferred to polyvinylidene fluoride membranes (Millipore, Billerica, MA, USA) at 20 V for 1 hr. After transfer, the membranes were then blocked with TBST (TBS containing 0.1% Tween 20) containing 5% non-fat milk at room temperature for 90 min. After blocking, the membranes were incubated with rabbit monoclonal anti-FMNL2 antibody (1:500), rabbit monoclonal anti-Arp2 antibody (1:500), mouse monoclonal anti-profilin1 antibody (1:500), rabbit monoclonal anti-Fascin antibody (1:5000), rabbit polyclonal anti-INF2 antibody (1:500), rabbit monoclonal anti-Grp78 antibody (1:1000), rabbit monoclonal anti-cofilin antibody (1:2000), rabbit

monoclonal anti-CHOP antibody (1:1000), or rabbit monoclonal anti-tubulin antibody (1:2000) at 4°C overnight. After washing five times in TBST (5 min each), membranes were incubated for 1 hr at room temperature with Horseradish Peroxidase (HRP)-conjugated Pierce Goat anti-Rabbit IgG (1:5000) or Horseradish Peroxidase-conjugated Pierce Goat anti-mouse IgG (1:5000). After washing for five times, the membranes were visualized using chemiluminescence reagent (Millipore, Billerica, MA). Every experiment repeated at least three times with different samples.

## Fluorescence intensity analysis

Immunofluorescence experiments were conducted simultaneously and with consistent parameters in both the control and treatment groups. The images were consistently captured using identical confocal microscope settings. Subsequently, the average intensity of fluorescence per unit area in the designated region of interest was quantified following the fluorescence staining. The acquired fluorescence data were analyzed employing *Zhu et al., 2011* and ImageJ software.

## Statistical analysis

All statistical analyses were performed using GraphPad Prism7.00 software (GraphPad, CA, USA), employing the *t*-test to assess the statistical significance between the control and treatment groups. The results were represented as the mean ± standard error of the mean. Statistical significance was defined as a p-value <0.05, denoted as *, ** for p < 0.01, *** and **** for p < 0.001 and p < 0.0001, respectively. The *n* represents the number of oocytes. Every experiment was conducted with a minimum of three biological replicates.

## Acknowledgements

We are particularly grateful to Xiao-Yan Fan and Xing-Hua Wang from Fertility Preservation Laboratory, Reproductive Medicine Center, Guangdong Second Provincial General Hospital for their technical assistance of live-cell imaging system. This work was supported by the National Key Research and Development Program of China (2023YFD1300502); the Fundamental Research Funds for the Central Universities of China (KYT2023002); the National Natural Science Foundation of China (32170857).

## Additional information

### Funding

| Funder | Grant reference number | Author |
| --- | --- | --- |
| National Key Research and Development Program of China | 2023YFD1300502 | Shao-Chen Sun |
| National Natural Science Foundation of China | 32170857 | Shao-Chen Sun |
| Fundamental Research Funds for the Central Universities | 2023YFD1300502 | Shao-Chen Sun |
| Fundamental Research Funds for the Central Universities | KYT2023002 | Shao-Chen Sun |
| Fundamental Research Funds for the Central Universities | 32170857 | Shao-Chen Sun |

The funders had no role in study design, data collection, and interpretation, or the decision to submit the work for publication.

### Author contributions

Meng-Hao Pan, Conceptualization, Data curation, Investigation, Methodology, Writing - original draft; Kun-Huan Zhang, Resources, Data curation, Investigation, Methodology; Si-Le Wu, Resources, Data

curation, Investigation; Zhen-Nan Pan, Resources, Data curation, Methodology; Ming-Hong Sun, Resources, Software, Methodology; Xiao-Han Li, Resources, Software, Formal analysis, Methodology; Jia-Qian Ju, Resources, Software; Shi-Ming Luo, Resources, Data curation, Formal analysis, Methodology; Xiang-Hong Ou, Resources, Software, Formal analysis; Shao-Chen Sun, Conceptualization, Supervision, Funding acquisition, Writing – review and editing, Project administration

## Author ORCIDs
Meng-Hao Pan ⓘ http://orcid.org/0000-0003-2120-9754
Shao-Chen Sun ⓘ http://orcid.org/0000-0001-5060-1742

## Ethics
We followed the guidelines of Animal Research Institute Committee of Nanjing Agricultural University to conduct the operations. The animal facility had license authorized by the experimental animal committee of Jiangsu Province (SYXK-Su-20170007).

Reviewer #1 (Public Review): https://doi.org/10.7554/eLife.92732.3.sa1
Reviewer #2 (Public Review): https://doi.org/10.7554/eLife.92732.3.sa2
Author response https://doi.org/10.7554/eLife.92732.3.sa3

## Additional files

### Supplementary files
• MDAR checklist

### Data availability
All data generated or analyzed during this study are included in the manuscript and supporting files; source data files have been provided for the mass spectrometry data and all the original images of blots from Figures 1, 2, and 4–6.

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
