## [Editor Report · eLife assessment]

This study presents **useful** findings regarding the role of formin-like 2 in mouse oocyte meiosis. Some of the data are supported by **incomplete** methodological details and analyses, and several conclusions are overstated. This paper would be of interest to reproductive biologists.

---

## [Referee Report · Reviewer #1 (Public Review)]

Summary:

The presented study focuses on the role of formin-like 2 (FMNL2) in oocyte meiosis. The authors assessed FMNL2 expression and localization in different meiotic stages and subsequently, by using siRNA, investigated the role of FMNL2 in spindle migration, polar body extrusion, and distribution of mitochondria and endoplasmic reticulum (ER) in mouse oocytes.

Strengths:

Novelty in assessing the role of formin-like 2 in oocyte meiosis

Weaknesses:

Overstating some of the presented data

Unconvincing analysis of the endoplasmic reticulum and mitochondria distribution

The authors addressed all my comments. The section materials and methods was improved. However, some statements still need to be clarified, as they seem to be overstated. I'm still not convinced about the main findings. For example, the analysis of ER and mitochondria distribution was based on a subjective assessment of clustering in meiosis I oocytes, and it's missing objective parameters and timing of the analysis.

Comments on revised version:

The authors addressed all my comments. The section materials and methods was improved. However, some statements still need to be clarified, as they seem to be overstated.

---

## [Referee Report · Reviewer #2 (Public Review)]

Summary:

This research involves conducting experiments to determine the role of Fmnl2 during oocyte meiosis I.

Strengths:

Identifying the role of Fmnl2 during oocyte meiosis I is significant.

Weaknesses:

The quantitative analysis and the used approach to perturb FMNL2 function would benefit from more confirmatory approaches and rigorous analysis.

Comments on revised version:

The authors addressed most of my comments. However, some comments were not addressed convincingly.

My concern is still valid. The authors used only one approach to knockdown FMNL2 which is "siRNA-mediated knockdown". Using an additional approach to inhibit FMNL2 (Trim-Away or morpholino,..) would be beneficial to confirm that the effect of siRNA-mediated knockdown of FMNL2 is specific.

Response 1: In the author's response, they mentioned that successful migration was quantified based on the contact between the spindle pole and the oocyte cortex.

After spindle migration, it is very common for the spindle to be close to (but not in contact with) the cortex for a considerable time. The spindle pole comes in contact with the cortex later (just before anaphase onset and polar body extrusion). Fig. 3A shows an example where at 9 h, the spindle is already migrated but did not come in contact with the cortex until 9:30 h. Based on Fig. 3B,C, the authors assessed spindle migration in fixed oocytes, making it impossible to fix all oocytes at the time of spindle contact with the cortex. Also,

the representative images in Fig. 3C do not show spindle staining to assess the contact between the spindle and the cortex.

Overall, I still believe that the distance between the spindle and the cortex is more accurate for quantifying spindle migration.

Response 2: The authors mentioned, "we made appropriate modifications to the relevant descriptions of immunoprecipitation experiments". I can't find these modifications in the manuscript. The authors need to state clearly that the immunoprecipitation results do not necessarily reflect meiotic oocytes specifically because these experiments were done using the whole ovary which contains both somatic cells and oocytes.

Response 5: The authors mentioned that "Based on our observations, during the extrusion of the first polar body in oocytes, there is a temporary occurrence of cellular morphological fragmentation due to cortical reorganization". Unfortunately, this means that the live imaging system in the authors' laboratory is not ideal for oocyte maturation. Several publications show normal oocyte morphology during cytokinesis. Please delete or replace Fig. 2E.

---

## [Author Response]

The following is the authors’ response to the original reviews.

**eLife assessment**
This study presents useful findings regarding the role of formin-like 2 in mouse oocyte meiosis. The submitted data are supported by incomplete analyses, and in some cases, the conclusions are overstated. If these concerns are addressed, this paper would be of interest to reproductive biologists.
**Public Reviews:**

**Reviewer #1 (Public Review):**
Summary:The presented study focuses on the role of formin-like 2 (FMNL2) in oocyte meiosis. The authors assessed FMNL2 expression and localization in different meiotic stages and subsequently, by using siRNA, investigated the role of FMNL2 in spindle migration, polar body extrusion, and distribution of mitochondria and endoplasmic reticulum (ER) in mouse oocytes.Strengths:Novelty in assessing the role of formin-like 2 in oocyte meiosis.Weaknesses:Methods are not properly described.Overstating presented data.It is not clear what statistical tests were used.My main concern is that there are missing important details of how particular experiments and analyses were done. The material and methods section are not written in the way that presented experiments could be repeated - it is missing basic information (e.g., used mouse strain, timepoints of oocytes harvest for particular experiments, used culture media, image acquisition parameters, etc.). Some of the presented data are overstated and incorrectly interpreted. It is not clear to me how the analysis of ER and mitochondria distribution was done, which is an important part of the presented data interpretation. I'm also missing important information about the timing of particular stages of assessed oocytes because the localization of both ER and mitochondria differs at different stages of oocyte meiosis. The data interpretation needs to be justified by proper analysis based on valid parameters, as there is considerable variability in the ER and mitochondria structure and localization across oocytes based on their overall quality and stage.

Thank you for your comment. We regret the oversight of omitting critical information in the manuscript. In the revised manuscript, we have included essential details such as mouse strains, culture media, stages of oocyte and statistical methods in the materials and methods section. Please find our details responses in the “Recommendations for the authors” part.

**Reviewer #2 (Public Review):**
Summary:This research involves conducting experiments to determine the role of Fmnl2 during oocyte meiosis I.Strengths:Identifying the role of Fmnl2 during oocyte meiosis I is significant.Weaknesses:The quantitative analysis and the used approach to perturb FMNL2 function are currently incomplete and would benefit from more confirmatory approaches and rigorous analysis.(1) Most of the results are expected. The new finding here is that FMNL2 regulates cytoplasmic F-actin in mouse oocytes, which is also expected given the role of FMNL2 in other cell types. Given that FMNL2 regulates cytoplasmic F-actin, it is very expected to see all the observed phenotypes. It is already established that F-actin is required for spindle migration to the oocyte cortex, extruding a small polar body and normal organelle distribution and functions.

Thank you for your comment. In the recent decade, Arp2/3 complex (Nat Cell Biol 2011), Formin2 (Nat Cell Biol 2002, Nat Commun 2020), and Spire (Curr Biol 2011) were reported to be 3 key factors to involve into this process. These factors regulate actin filaments in different ways. However, how they cross with each other for the subcellular events were still fully clear. Our current study identified that FMNL2 played a critical role in coordinating these molecules for actin assembly in oocytes. Our findings demonstrate that FMNL2 interacts with both the Arp2/3 complex and Formin2 to facilitate actin-based meiotic spindle migration. Additionally, we discovered a novel role for FMNL2 in determining the distribution and function of the endoplasmic reticulum and mitochondria, which may in turn influence meiotic spindle migration in oocytes. Our results not only uncover the novel functions of FMNL2-mediated actin for organelle distribution, but also extend our understanding of the molecular basis for the unique meiotic spindle migration in oocyte meiosis.

(2) The authors used Fmnl2 cRNA to rescue the effect of siRNA-mediated knockdown of Fmnl2. It is not clear how this works. It is expected that the siRNA will also target the exogenous cRNA construct (which should have the same sequence as endogenous Fmnl2) especially when both of them were injected at the same time. Is this construct mutated to be resistant to the siRNA?

Thank you for your question. We regret any misunderstanding that may have been caused by the inappropriate description in our manuscript. In the rescue experiments, we initially injected FMNL2 siRNA into oocytes, followed by the microinjection of FMNL2 mRNA 18-20 hours later. After conducting our previous experiments, we have verified through Western blotting that endogenous FMNL2 is effectively suppressed 18-20 hours following the microinjection of FMNL2 siRNA. Additionally, we observed a significant increase in exogenous FMNL2 protein expression 2 hours after the injection of FMNL2 mRNA. We believe that the exogenous FMNL2 could compensate the decrease by FMNL2 knockdown, and this approach was adopted in many oocyte studies.

(3) The authors used only one approach to knockdown FMNL2 which is by siRNA. Using an additional approach to inhibit FMNL2 would be beneficial to confirm that the effect of siRNA-mediated knockdown of FMNL2 is specific.

Thank you for your question. Yes, the specificity is always the concern for siRNA or morpholino microinjection due to the off-target issue. Due to the limitation we could not generate the knock out model, and there are no known inhibitors with specific targeting capabilities for FMNL2. To solve this, we performed the rescue study with exogenous mRNA to confirm the effective knock down of FMNL2. These measures provide reassurance regarding the credibility of the experimental outcomes, and this is also the general way to avoid the off-target of siRNA or morpholino.

**Reviewer #3 (Public Review):**
Summary:The authors focus on the role of formin-like protein 2 in the mouse oocyte, which could play an important role in actin filament dynamics. The cytoskeleton is known to influence a number of cellular processes from transcription to cytokinesis. The results show that downregulation of FMNL2 affects spindle migration with resulting abnormalities in cytokinesis in oocyte meiosis I.Weaknesses:The overall description of methods and figures is overall dismissively poor. The description of the sample types and number of replicate experiments is impossible to interpret throughout, and the quantitative analysis methods are not adequately described. The number of data points presented is unconvincing and unlikely to support the conclusions. On the basis of the data presented, the conclusions appear to be preliminary, overstated, and therefore unconvincing.

Thank you for your comment. We regret the oversight of omitting critical information in the manuscript. In the revised manuscript, we have incorporated your suggestions for modification, particularly regarding the Materials and Methods section. Please see the detailed revision and responses in the “Recommendations for the authors” part.

**Recommendations for the authors:**

**Reviewer #1 (Recommendations for The Authors):**
My main concern is that there are missing important details of how particular experiments and analyses were done. The material and methods section is not written in the way that presented experiments could be repeated - it is missing basic information (e.g., used mouse strain, timepoints of oocytes harvest for particular experiments, used culture media, image acquisition parameters, etc.). Some of the presented data are overstated and incorrectly interpreted. It is not clear to me how the analysis of ER and mitochondria distribution was done, which is an important part of the presented data interpretation. I'm also missing important information about the timing of particular stages of assessed oocytes because the localization of both ER and mitochondria differs at different stages of oocyte meiosis. The data interpretation needs to be justified by proper analysis based on valid parameters, as there is considerable variability in the ER and mitochondria structure and localization across oocytes based on their overall quality and stage. My specific comments are listed below.(1) Information about statistical tests that were used needs to be provided for all quantification experiments.

Thank you for your suggestion. Based on your suggestions, we revised the statistical analysis description in the Materials and Methods section. Additionally, we also included a description of the statistical methods in the legends of the relevant result figures.

(2) I recommend replacing the plunger plots, used in most quantification data, with alternatives allowing evaluation of the distribution of the data (dot plots, box plots, whisker plots).

Thank you for your suggestion. Following your suggestion, we replaced the plunger plots in Fig 2C, D, H, I and Fig3 B, C with dot plots.

(3) Can the authors provide information about particular time points when were individual oocyte stages (GVBD, meiosis I, and meiosis II) harvested/used for immunofluorescence protein detection, western blotting, microinjection, and ER and mitochondria staining? Were the time points always the same in all presented experiments and experimental vs control group? If not, this needs to be clarified.

Thank you for your suggestion. We used oocytes in the metaphase I (MI) stage for the statistical analysis of spindle migration, actin filament aggregation, endoplasmic reticulum localization, and mitochondrial localization. In the Western blot analysis, GV stage oocytes were utilized to evaluate the efficiency of knockdown and rescue experiments. The protein expression levels of Arp2, Formin2, INF2, Cofilin, Grp78, and Chop in different treatment groups were detected using MI-stage oocytes. In the revised version, we provided all the detailed information about the stages.

(4) Figure 1B: Can the authors comment on why there is a missing representative image of MII oocyte FMBL2-Ab? I recommend including this in the figure to have a complete view of comparing overexpressed and endogenous FMNL2 localization in oocyte meiosis.

Thank you for your suggestion. In the revised manuscript, we added immunostaining images of FMNL2 antibody in MII stage oocytes.

(5) Figure 1C: The figure legend says, "FMNL2 and actin overlapped in cortex and spindle surrounding". In MI oocytes, there is usually no accumulated actin signal around the spindle, which is also true in the presented images, so there cannot be overlapping with the FMNL2 signal. The interpretation should be changed.

We apologize for this inappropriate description that was used, and we deleted this sentence.

(6) Figure 2B: What were the parameters of the "large" and "normal" polar bodies for performing the analysis?

Thank you for your question. In order to assess the size of the polar body, we conducted a comparison between the diameter of the polar body and that of the oocyte. If the diameter of the polar body was found to be less than 1/3 of the oocyte's diameter, we categorized it as normal-sized polar body. Conversely, if the polar body's diameter exceeded 1/3 of the oocyte's diameter, we categorized it as a large polar body. We have included these details in the Results section of the manuscript.

(7) Figure 2F: Can the authors comment on what can be the second band in the rescue group?

Thank you for your question. In the rescue experiment, we microinjected exogenous FMNL2-EGFP mRNA into the oocytes. As a result, compared to endogenous FMNL2, the protein size increased due to the addition of the EGFP tag, approximately 27 kDa. Hence, in the Western blot bands of the rescue group, the upper band represents the expression of exogenous FMNL2-EGFP, while the lower band corresponds to the expression of endogenous FMNL2. We have provided annotations in the revised Figure 2F to clarify this.

(8) Can the authors comment on the variability of PBE between 2C and 2H in the FMNL2-KD groups? In panel C, the PBE in the KD group was 59.5 {plus minus} 2.82%; in panel H, the PBE in the KD group was 48.34 {plus minus} 4.2%, and in the rescue group, the PBE was 62.62 {plus minus} 3.6%. The rescue group has a similar PBE rate as the KD group in panel C. How consistent was the FMNL2 knockdown across individual replicates? Can the authors provide more details on how the rescue experiment was performed?

Thank you for your question. We believe that the difference in PBE observed in Figure 2C and 2H of the FMNL2-KD group was due to the microinjection times and the duration of in vitro arrest. The results shown in Figure 2C depict the outcome of a single injection of FMNL2 siRNA into GV stage oocytes, followed by 18 hours of in vitro arrest; the results shown in Figure 2H contain a subsequent additional injection of FMNL2-EGFP mRNA with another 2 hours of arrest. The two rounds of microinjection and the extended period of in vitro arrest both affect oocyte maturation rates.

(9). Figure 2J and K: What groups were compared together? The used statistic needs to be properly described.

Thank you for your question. The FMNL2-KD, FMNL3-KD, and FMNL2+3-KD groups were all compared to the Control group, therefore, t-test was used for analysis. We have provided explanations in the revised manuscript.

(10) Figure 4B and C: Can the authors provide representative images without oversaturated actine signal?

Thank you for your question. For the analysis of oocyte F-actin, the F-actin are divided into cortex actin and cytoplasmic actin. Due to the contrast during imaging, the strong cortex actin signals affected the detection of cytoplasmic actin, therefore, it is necessary to increase the scanning index, which will cause the overexpose the cortex actin signal. This is for the better observation of the cytoplasmic signals.

(11) Figure 4G + 5H: Can the authors comment on why they used as a housekeeping gene actin instead of tubulin, which was used in the rest of the WB experiments?

Thank you for your question. In most of the western blot experiments conducted in this study, we used tubulin as a housekeeping gene. However, due to the supply of antibodies by delivery period, we had GAPDH and actin as well for some experiments. These housekeeping genes were all valid for the study.

(12) Based on what parameters was ER considered normally or abnormally distributed, and what stages of oocytes were assessed?

Thank you for your question. In this study, we employed oocytes at the MI stage for the analysis of ER localization. In the MI stage, the ER localized around the spindle, which is regarded as the typical localization pattern. The ER displayed a dispersed distribution throughout the cytoplasm or clustered were categorized as aberrant positioning. We included relevant descriptions in the revised version of the manuscript.

(13) Figure 5H: As a housekeeping gene was used actin - the quantification is labeled as a Grp78 to tubulin ratio.

Thank you for pointing out the error. This is a label mistake and we corrected it.

(14) Information about how JC-1 staining was done needs to be provided.

Thank you for your carefully reading. We included a description of JC1 staining in the Materials and Methods section.

(15). Line 231-232: "As shown in Figure 4A" - the text doesn't correspond to the figure.

Thank you for pointing out the error. We revised this mistake in the revised manuscript by correcting "Fig3A" to "Fig4A."

(16) Line 265: there is probably a missing word "Formin2".

Thank you and we corrected the error and made the necessary changes in the revised manuscript.

**Reviewer #2 (Recommendations for The Authors):**
(1) Quantification and analysis:Fig. 3B: The rate of spindle migration should be quantified based on the distance from the spindle to the cortex. Also, the orientation of the spindle (Z-position) needs to be taken into consideration.Fig. 5C, D: It is unclear how the rate of ER distribution was calculated.Western blot: In many experiments (such as Fig. 5H), the bands are saturated which will prevent accurate intensity measurements and quantifications.

For spindle migration, we specifically focused on spindles exhibiting a distinctive spindle-like shape with clear bipolarity to eliminate any statistical discrepancies potentially caused by variations in Z-axis alignment. Our criterion for determining successful migration was based on the contact between the spindle pole and the cortical region of the oocyte. Therefore, we think that the rate is better to reflect the phenotype than the distance.

For the examination of ER localization, Reviewer 1 also raised this issue. We utilized oocytes at the MI stage in this study. The ER localized around the spindle in MI stage. The ER displayed a dispersed distribution throughout the cytoplasm or clustered were categorized as aberrant positioning. We included relevant descriptions in the revised version of the manuscript.

For the bands of the western blot results, during the experimental procedure we typically capture multiple images at different exposure levels (3-5 images). In the revised manuscript, we have replaced the inappropriate images with more suitable ones.

(2) Given that all Immunoprecipitation experiments in this manuscript were performed on the whole ovary which contains more somatic cells than oocytes, the results do not necessarily reflect meiotic oocytes. Please consider this possibility during the interpretation.

Thank you for your suggestion. Yes, we agree with you. In the revised manuscript, we made appropriate modifications to the relevant descriptions.

(3) 351-365: The conclusion that Arp2/3 compensates for the decreased formin 2 in FMNL2 knockdown oocytes is a bit unconvincing. 1- In mouse oocytes, it is already known that Arp2/3 and formin 2 regulate different pools of F-actin nucleation. 2- The authors found an increase in Arp2/3 in FMNL2 knockdown oocytes compared to control oocytes without any change in cortical F-actin. Given that Arp2/3 is primarily promoting cortical F-actin, it is expected to see an increase in cortical F-actin in FMNL2 knockdown oocytes, which was not the case.

Thank you for your question. Yes, previous studies showed that formin2 localizes to the cytoplasm of oocytes and accumulates around the spindle, which facilitate cytoplasmic actin assembly. While Arp2/3 is primarily responsible for actin assembly at the cortex region of oocytes. In invasive cells, FMNL2 is mainly localized in the leading edge of the cell, lamellipodia and filopodia tips, to improve cell migration ability by actin-based manner (Curr Biol 2012). We showed that FMNL2 localized both at spindle periphery and cortex, but depletion of FMNL2 did not affect cortex actin intensity. We think that FMNL2 and Arp2/3 both contribute to the cortex actin dynamics, when FMNL2 decreased, ARP2 increased to compensate for this, which maintained the cortex actin level. In the revised manuscript, we have made modifications to avoid excessive extrapolation from our results, ensuring that our conclusions are presented in a more objective manner.

(4) Lines 195-197: The spindle is initially formed soon after the GVBD, so there is no spindle during GVBD. Also, I can't see oocytes at anaphase I or telophase I in this figure. Please revise.

Thank you for your suggestion. We apologize for the inappropriate descriptions that were used. In the revised manuscript, we have made modifications to the respective descriptions in the Results part.

(5) Fig. 2E: It seems that the control oocyte is abnormal with mild cytokinesis defects. Please replace or delete it since this information is already included in Fig. 3A.

Thank you for your suggestion. Based on our observations, during the extrusion of the first polar body in oocytes, there is a temporary occurrence of cellular morphological fragmentation due to cortical reorganization (11h in control oocyte from Fig 2E). However, after the extrusion of the first polar body, the oocyte morphology returns to normal. Figure 2E illustrates the meiotic division process of oocytes, while Figure 3A primarily focuses on the process of oocyte spindle migration. We think that it is better to retain both to present our results.

**Reviewer #3 (Recommendations for The Authors):**
In the case of the observed phenotype, the stage of GV is important. The phenotypes presented also occur in meiotic or developmentally incompetent oocytes. In addition, the images of GV oocytes appear as NSN, which also show the KD phenotype in Figs. 2 and 3.

Thank you for your concern. As the oocyte grows, the proportion of SN-type oocytes gradually increases. When the oocyte diameter reaches 70-80 μm, the proportion of SN oocytes is approximately 52.7% (Mol Reprod Dev. 1995). In our study, both the control and knockdown groups collected oocytes with a diameter of around 80 μm, which is considered as fully-grown oocytes, predominantly in the SN phase. Since the collection period and size of the oocytes were consistent, we can sure that the observed differences between the control and knockdown groups in phenotype analysis could be solid and reliable.

MII is absent in Fig. 1B.

In the revised manuscript, we added immunostaining images of FMNL2 in MII stage oocytes.

The result of KD is not convincing. Also, discuss whether the heterozygous effect of Fmnl2 deletion affects reproductive fitness.

Thank you for your concern. In our investigation, limited to the setup of knock out model, we employed siRNA to knockdown FMNL2 expression, to avoid the risk of off-target, we performed rescue experiment with exogenous mRNA, which we believe that it could solve this issue. When designing siRNA sequences, we ensured their specificity for binding to FMNL2 mRNA only, and we assessed the levels of FMNL2 and FMNL3 mRNA in oocytes after injection of FMNL2 siRNA. The results showed that, compared to the control group, the expression of FMNL2 mRNA decreased by approximately 70% after 18 hours of FMNL2 siRNA injection, while the level of FMNL3 mRNA was not decreased.

Fig. 2F rescue experiment with double bands. What bands are seen here? Did the authors inject tagged or untagged FMNL2? Or does endogenous FMNL2 appear higher in the sample after KD?

Thank you for your question. In the rescue experiment, we microinjected exogenous FMNL2-EGFP mRNA into the oocytes. As a result, compared to endogenous FMNL2, the protein size increased due to the addition of the EGFP tag, approximately 27 kDa. Hence, in the Western blot bands of the rescue group, the upper band represents the expression of exogenous FMNL2-EGFP, while the lower band corresponds to the expression of endogenous FMNL2. We provided annotations in the revised Figure 2F to clarify this.

Variability in mitochondria and ER distribution patterns is also known in healthy and developing oocytes, although the authors described only a single phenotype.

Thank you for your concern. Yes, mitochondria and ER show dynamic localization in different stage of oocyte maturation. However, in this study we employed oocyte MI stage for the analysis of ER and mitochondria localization, and in MI stage, both the ER and mitochondria localize around the spindle. This pattern is considered as the normal localization. Several studies showed that dispersed or clustered localization contributed to maturation defects. We included relevant descriptions in the revised manuscript.

What exactly is meant by input in the IP experiments? Why is the target missing in the input sample?

Thank you for your question. We subjected the input samples to electrophoresis on a single channel, all the analyzed proteins demonstrated normal expression, thereby confirming the viability of the input sample. However, upon simultaneous exposure with the IP samples, we observed a lack of clear signal for certain proteins in the input group. This phenomenon is due to the excessive signal intensity resulting from protein enrichment in the IP group, which caused the low exposure of proteins in input group.

Explain the rationale for using, actin or tubulin as loading or normalization controls in the study focusing on the cytoskeleton.

Thank you for your question. Actin and tubulin are both widely used as the control due to their stable expression. For actin, there are α-actin and β-actin isoforms. Formins and Arp2/3 complex regulate the polymerization of α-actin and β-actin to form F-actin, not isoform expression. In our study F-actin (the functional type) was examined. While α-tubulin and β-tubulin are two subtypes of tubulin, and they interact with each other to form stable α/β-tubulin heterodimers. The changes of cytoskeleton dynamics could not change the expression of α/β-tubulin. Therefore, β-actin and α-tubulin could be used as normalization controls.

Fig. 6E shows only *, but the legend says **.

Thank you for pointing out the error. We correct the mistake in the revised manuscript.

Spindle positioning appears to differ between control and KD. Does this affect the quantification of Fig. 6F? Adequate nomenclature should be used here.

Thank you for your question. Yes, spindle positioning was affected by FMNL2 depletion. However, central spindle or cortex spindle all belong to MI stage, and JC1 is not related with the stage difference. To avoid misunderstanding we replaced the representative images and corresponding description in Figure 6F.

The description of the methods and legends should be significantly improved.

Thank you for your suggestion. Reviewer 1 and 2 also raised the similar concern. We enriched the description of methods and legends in the revised manuscript.